# Characterisation of oxygen defects and nitrogen impurities in TiO$_2$ photocatalysts using variable-temperature X-ray powder diffraction

Christopher Foo [1,2], Yiyang Li[1], Konstantin Lebedev[1], Tianyi Chen[1], Sarah Day[2], Chiu Tang[2] & Shik Chi Edman Tsang [1✉]

TiO$_2$-based powder materials have been widely studied as efficient photocatalysts for water splitting due to their low cost, photo-responsivity, earthly abundance, chemical and thermal stability, etc. In particular, the recent breakthrough of nitrogen-doped TiO$_2$, which enhances the presence of structural defects and dopant impurities at elevated temperatures, exhibits an impressive visible-light absorption for photocatalytic activity. Although their electronic and optical properties have been extensively studied, the structure-activity relationship and photocatalytic mechanism remain ambiguous. Herein, we report an in-depth structural study of rutile, anatase and mixed phases (commercial P25) with and without nitrogen-doping by variable-temperature synchrotron X-ray powder diffraction. We report that an unusual anisotropic thermal expansion of the anatase phase can reveal the intimate relationship between sub-surface oxygen vacancies, nitrogen-doping level and photocatalytic activity. For highly doped anatase, a new cubic titanium oxynitride phase is also identified which provides important information on the fundamental shift in absorption wavelength, leading to excellent photocatalysis using visible light.

[1] Wolfson Catalysis Centre, Department of Chemistry University of Oxford, Oxford, UK. [2] Diamond Light Source, Didcot, UK. ✉email: edman.tsang@univ.ox.ac.uk

TiO$_2$-based materials are widely studied as efficient photo-catalysts since the first photoelectrochemical (PEC) water-splitting system was demonstrated in 1972. They have found use in a wide range of applications including the photo-catalytic hydrogen evolution reaction, pollutant degradation[1,2], and dye-sensitised photocatalysis[3], due to their characteristic strong photo-responsive, low cost, high natural abundance, and high chemical and thermal stability.

There are many polymorphs of TiO$_2$, including anatase (A), rutile (R) and brookite, which all exist naturally in minerals. There is a consensus from experimental evidence that anatase is the most active for photocatalysis, there is no in-depth under-standing of its structure-activity relationship. In addition, the wide bandgap of TiO$_2$ anatase limits its absorption of visible light in photocatalysis, which has driven the development of many modifications of TiO$_2$-based photocatalyst materials. Cation-doping or anion-doping of the oxide is presently regarded as a highly effective approach for enhancing the utilisation of visible light[4–7], among which nitrogen-doping has received significant attention as it reduces the band gap for photo-excitation, enabling the absorption of the plentiful visible-light region of the solar-irradiance spectrum, thus resulting in enhanced photocatalytic performance. The electronic factors that contribute to the pho-tocatalytic activity of TiO$_2$, with and without doping, have been researched for some years, but the underlying mechanism is still not yet clear. Generally, it is thought that the extraction of neutral oxygen atoms, results in electron transfer to the Ti cationic band structure resulting in n-type doping[8], which may occur in both the bulk and on the surface of TiO$_2$ crystals. However, photo-catalytic water splitting is a surface phenomenon that takes place by the direct interaction of water molecules with the catalyst. Thus, only the excited Ti sites and holes that are on the surface are directly involved in the processes of proton reduction and hydroxide oxidation, which form hydrogen and oxygen gas, respectively, from dissociated water. Nonetheless, there is still a lack of detail with respect to the structure-activity relationship of this unique phase-boundary, and the exact way in surface and bulk structure affects the system is still far from well-understood. Early reports of TiO$_2$-based photocatalysts concede that the enhanced photocatalytic activity cannot be directly correlated to changes in bulk structure by crystallographic methods, as such small changes cannot be easily resolved[4]. On the other hand, the formation of defects during photocatalysis, such as interstitial ions and vacancies with unpaired electrons, can be observed by EPR measurements[6,9]. The surface defects in the photo-catalytically active anatase (101) facet have been visualised by STEM, but the surface concentration is found to be too low at room temperature to account for photocatalysis[10]. On the other hand, total defects, that is to say irrespective of depth (surface, sub-surface and bulk), may be characterised by resonant photo-emission spectroscopy (RESPES) and X-ray absorption spectro-scopy (XAS). It has been shown that the polymorph and exposed facets of TiO$_2$ materials significantly affect the concentration of total oxygen vacancies[11]. Proportionate to photocatalytic activity, the anatase (101) surface supports a very high bulk concentration, which indicates oxygen vacancies beyond the surface; meanwhile rutile (110) and anatase (001) have very low concentrations of total vacancies. This finding was further supported by inter-rogating the energies of various oxygen vacancy sites by first-principles DFT calculations[12]. Though these reports all highlight the nature and importance of surface structural change of TiO$_2$ powder, they make no reference to the parent bulk structure, which along with an atomic understanding of structure, is the key to understanding surface photocatalysis.

TiO$_2$ structures also display rich phase-variation across tem-perature and pressure, adopting many common mineral structures[13]: monoclinic $Pbmm$ tungsten bronze (TiO$_2$(B),)[14], monoclinic $P2_1/c$ ZrO$_2$ baddeleyite (TiO$_2$(MI),)[15], orthorhombic $Pbcn$ columbite (TiO$_2$(OI))[16], orthorhombic $Pnma$ PbCl$_2$ cotunnite TiO$_2$(OII)[17], cubic $Fm\bar{3}m$ CaF$_2$ (TiO$_2$(C1))[18–20]. In addition, orthorhombic $Pbnm$ MnO$_2$ ramsdellite (TiO$_2$(R))[21] and monoclinic $I2/m$ hollandite (TiO$_2$(H))[22] structures can also be formed by the complete oxidation of lithium-doped and potassium-doped titania, respectively.

Treating anatase powder with ammonia at elevated tempera-ture has been experimentally shown to yield full conversion of tetragonal anatase to cubic titanium nitride, which has itself been researched as a photocatalyst, particularly due to its broad absorption in the visible region[23–25]. Titanium nitride consists of titanium in an oxidation state of +3 and nitrogen in an oxidation state of −3, arranged in the $Fm\bar{3}m$ rock-salt structure. The material is widely recognised as a material for bio-electrical components[26]. Thin films can be synthesised using solid-liquid solvothermal conditions[27], IVD (ion vapour deposition) or CVD (chemical vapour deposition)[28]. Instead of total nitridation of TiO$_2$, there are several reports of the formation of titanium oxynitride for application in photocatalytic and photonic mate-rials[29]. However, though phase identification for complex poly-phasic TiO$_2$ powders yielded by ammonia/argon treatment have been reported[30,31], the structures of titanium oxynitride remain largely inferred and uncharacterised. Do et al. report the tuneable non-stoichiometric composition of titanium oxynitride, TiN$_x$O$_y$, between the end-phases of pure TiO$_2$ and TiN[32]. Across reports, it is generally noted that these intermediate phases have an intermediate lattice parameter between the two end-composition $Fm\bar{3}m$ phases of titanium(II) oxide (4.185 Å) and titanium nitride (4.241 Å)[27,32]. Some have used this observation to apply the linear Vegard's law and assume substitutional doping to yield values for composition[27]. Despite this, there are no complete structure solutions for the oxygen site within the nitride structure so far. In addition, the nitrogen-doping promotes the formation of oxygen vacancies in TiO$_2$[7,33]. We have previously reported that oxygen vacancies formed during nitrogen-doping at elevated temperature greatly contribute to improved visible-light absorption and act as the surface active sites, leading to enhanced photocatalytic water splitting activities and quantum efficiencies[34]. However, the in-depth structural characterisations of the doped nitrogen atoms and oxygen vacancies, and their relationship to the photocatalytic mechanism are still urgently required to inform further catalyst development and fundamental understanding.

X-ray diffraction is a well-known technique for monitoring changes in crystalline structures, but often suffers from a lack of chemical hypothesis related to photocatalysis, and in particular interrogation of the effect of defects and impurities on the deri-vative of lattice parameter against temperature[35,36]. Despite these structural investigations into the promotion of photocatalytic activity of TiO$_2$ based materials, it is imperatively important to gain understanding on the structural-activity relationship: namely which phases are relevant to the photocatalysis and the funda-mental underlying principles for the strong photocatalytic effect at different concentrations of impurities and at different temperatures.

Herein, we have presented a detailed study of the photo-catalytic activity of N–TiO$_2$ materials under visible light, and analysed structural changes using variable-temperature synchro-tron X-ray powder diffraction (VT-SXPD), showing bulk reor-ganisation at dynamic equilibrium as a result of sub-surface oxygen vacancy formation. We report that there is a significant change in the c-axis lattice expansion in anatase under elevated temperature, resulting in a unique non-linear expansion due to the Jahn–Teller effect. However, as surface oxygen vacancies are formed at temperatures above 200 °C, a concomitant decrease in

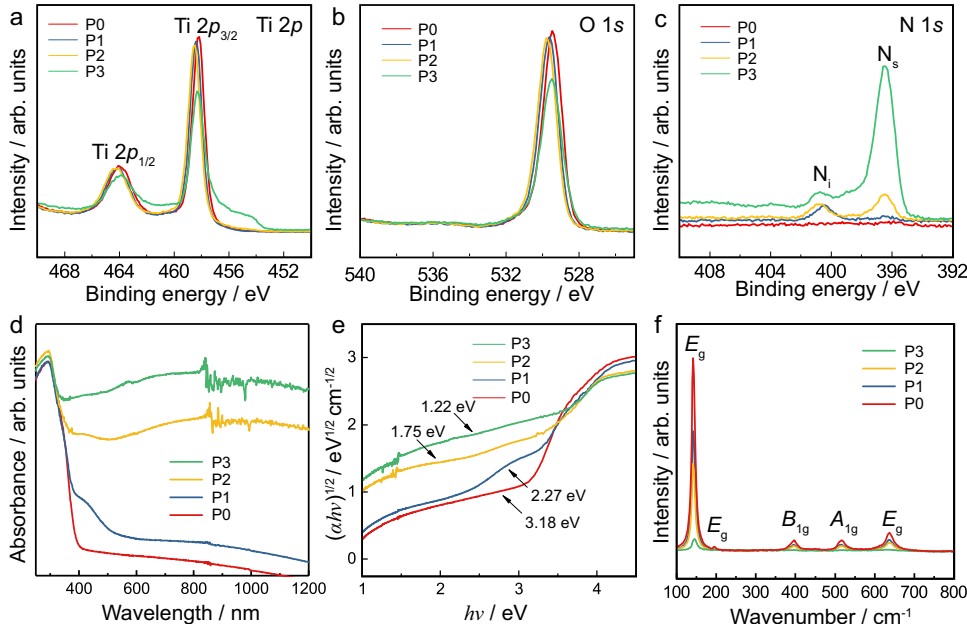

**Fig. 1 Electronic properties of N-doped P25.** XPS spectra of N-doped P25 TiO$_2$ treated with ammonia flow at different temperatures: (**a**) Ti 2$p$, (**b**) O 1$s$ and (**c**) N 1$s$. Photo-absorption properties characterised by (**d**) UV-vis absorption spectra, (**e**) corresponding Tauc plots and average band gaps with fitting errors obtained, and (**f**) Raman spectra.

**Table 1 Nitrogen concentration in P25 samples by XPS and ICP-MS; substitutional nitrogen (N$_s$) and interstitial nitrogen (N$_i$) in lattice.**

| Sample | NH$_3$ treatment temperature/°C | Nitrogen composition by XPS/%wt | N$_i$/% | N$_s$/% | N$_i$: N$_s$ ratio | Nitrogen composition by ICP-MS/%wt | Band gap by Tauc plot (E$_g$)/eV |
|---|---|---|---|---|---|---|---|
| P0 | N/A | 0 | N/A | N/A | N/A | 0 | 3.18 |
| P1 | 550 | 0.53 | 71.8 | 28.2 | 2.54 | 0.6 | 2.27 |
| P2 | 600 | 1.4 | 53.7 | 46.3 | 1.16 | 2.4 | 1.75 |
| P3 | 620 | 4.6 | 28.2 | 71.8 | 0.39 | 5.3 | 1.22 |

the *c*-axis expansion in lattice is observed, which we have shown to be correlated to photocatalytic activity. We demonstrate oxygen-vacancy mobility from the disordered surface to form ordered sub-surface vacancies, as resolved by SXPD. The technique clearly indicates that photocatalysis in anatase is greatly facilitated by large quantities of oxygen vacancies and Ti$^{3+}$ beyond the surface trilayer, particularly in comparison to the inactive rutile polymorph. The inclusion of nitrogen can clearly increase the quantity of sub-surface oxygen vacancies and stabilise the anatase phase as reflected by the different degrees of observed unit-cell distortion. By doping high levels of nitrogen into anatase, a new cubic titanium oxynitride phase is identified, which provides an additional component in the fundamental understanding of the bandgap modification in N–TiO$_2$ materials, and will inform wider hypotheses and perspectives on this system. Consequently, we present important links for the first time between structure, sub-surface oxygen vacancies, nitrogen-doping, and photocatalytic activity of anatase catalysts at various temperatures. Ultimately, we report the influence of well-characterised structural modifications on electronic phenomena in this photocatalytic hydrogen-evolution powder catalyst.

## Results and discussion
**Synthesis and XPS and EPR characterisation.** An initial study was carried out on the commercially available P25 TiO$_2$ powder, which consists of approximately 75%wt anatase and 25%wt rutile.

The degree of nitrogen-doping was controlled by treating the P25 in NH$_3$ flow at different temperatures. The analysis by TEM and HAADF-STEM micrographs indicates the successful formation of TiO$_2$-based nanoparticles (Fig. S2 and S3). Elemental analysis by ICP-MS shows there is a non-linear proportionate relationship between NH$_3$ treatment temperature and the nitrogen content at each temperature. The trend of increasing nitrogen content with temperature was also observed by XPS in the increasing N 1$s$ signal (Fig. 1). The nitrogen-composition values yielded by elemental analysis are consistently higher than those derived from XPS. The small disagreement is attributed to differences in the penetration depth of the techniques; XPS only probes the surface, whereas elemental analysis reflects total composition. Hence, the discrepancy indicates non-homogeneous elemental distribution with respect to depth, namely lower nitrogen composition at the surface. The titanium and oxygen signals were observed as expected. Two peaks corresponding to Ti 2$p_{3/2}$ and Ti 2$p_{1/2}$ were observed at the binding energies of 458 eV and 464 eV, respectively, which can be attributed to the main characteristic peaks of Ti$^{4+}$ in TiO$_2$ materials. The O 1$s$ peak at 530 eV is the characteristic peak of oxygen in the TiO$_2$ lattice and N-doped TiO$_2$. The N 1$s$ XPS spectra shows clearly two characteristic peaks, located at 396 eV and 401 eV, which can be assigned to substitutional N (N$_s$) on O sites, and interstitial N atoms in the lattice (N$_i$), respectively[6,9]. It is known that synthesis precursors, conditions, and stoichiometry have significant impact on the resulting N$_i$:N$_s$ ratio[37]. As summarised in Table 1, from pristine P0 to

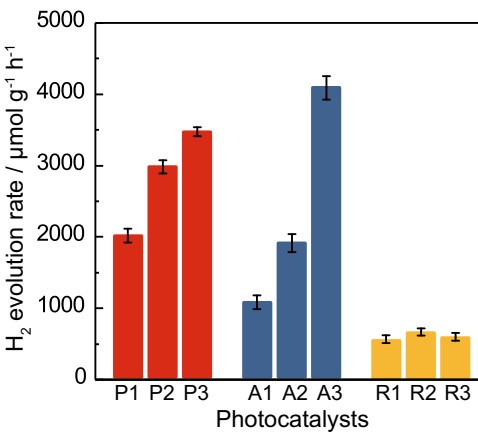

**Fig. 2 Photocatalytic water splitting activity study of the N-doped TiO$_2$ samples.** Photocatalytic water splitting activities over all the as-prepared N-doped TiO$_2$ samples at 270 °C under visible-light irradiation (red: P25; blue: anatase; yellow: rutile). Error bars indicate the standard deviation.

P1, ammonia treatment at (550 °C) is shown to result in 0.5%wt nitrogen-doping; this occurs primarily at the interstitial site ($N_i$: $N_s = 2.54$). For P2 (600 °C), the signal for substitutionally-doped nitrogen increases whilst the interstitial signal remains approximately unchanged ($N_i$:$N_s = 1.16$). The treatment temperature of 620 °C for P3 results in a significantly greater nitrogen composition of 4.6%wt, which significantly favours occupancy of substitutional sites ($N_i$:$N_s = 0.39$). This is in agreement with the literature, in that substitutional doping requires higher temperatures[38].

As has long been known, increasing the nitrogen content in anatase reduces the band gap, which is advantageous, in that the absorption profile is shifted to include more of the visible spectrum. We have shown this for P25 by UV-vis spectroscopy, in that in addition to the absorption edge at 390 nm, an additional broadened edge at 500 nm is observed for samples P1-3 (d). Higher temperature ammonia treatment leads to significant broadening, shifting the additional absorption edge to the visible and infrared regions. Tauc plots show that nitrogen-doping reduces the band gap from 3.18 eV to 1.22 eV (e), making it easier to form the active Ti$^{3+}$ state, and leading to greater utilisation of the solar spectrum. The photo-excitation of these colour centres and defects as extra intra-band levels to the conduction band can therefore contribute to the visible-light absorption of the N-doped TiO$_2$[5,7]. This wavelength of the absorption lengthens from 390 nm to 1000 nm, resulting in near-complete absorption of the visible spectrum (380–740 nm). P2 and P3 even show quite strong absorption in the near-IR region (700–1000 nm). This was also seen by a colour change from white pristine samples to the darker, grey of the intermediate _1 and _2 samples and black _3 samples. Five major peaks that represent $E_g$, $E_g$, $B_{1g}$, $A_{1g}$ and $E_g$ Raman-active vibrational modes are located at 144, 196, 396, 544 and 636 cm$^{-1}$, respectively (Fig. 2f), indicating that the predominant phase of the N-doped TiO$_2$ is anatase[39,40], which is consistent with the XRD results (Fig. S4). The peaks of N-doped samples exhibit weakening and broadening, due to decreasing crystallinity caused by both interstitial and substitutional nitrogen-dopant and the formation of oxygen vacancies[41,42].

**Study of photocatalytic water splitting at elevated temperature.** In order to obtain a clear understanding of the behaviours of different phases, the constituent P25 phases of anatase and rutile were treated in their pure phases with ammonia using the same

procedure. Catalytic testing of all samples was carried out at 270 °C. For the undoped samples, no activity was observed under visible light. Though when the irradiation spectrum was broadened to UV/visible light, the pristine samples were shown to be photocatalytically responsive to UV light (Fig. S5). This is in agreement with the photo-absorption studies of P25 materials above. However, even under UV/visible light, the activity of the non-doped samples were reasonably low (<600 μmol g$^{-1}$ h$^{-1}$). On the other hand, the N-doped catalysts are highly active under visible light only (Fig. 2). In order to show that this reaction is photocatalytic in nature, control tests were carried out, which used (1) no catalyst, (2) no UV or visible light (P3 catalyst used), and (3) no catalyst and no UV or visible light. These control tests yielded zero hydrogen evolution, clearly showing that both light and the N–TiO$_2$ catalyst are required for hydrogen evolution to occur. Furthermore, GC analysis was carried out over a multi-cycle test and a long duration test. Firstly, a P3 catalyst was reused over five 4-h batch reactions, and showed to yield H$_2$ and O$_2$ in a 2:1 ratio which is indicative of the stoichiometric splitting of water (Fig. S6). These observations were consistent over all cycles with no evidence of the stoichiometric consumption of the catalyst through decreasing yields. Similarly, we have also carried out a single 50-h test in which the stoichiometric H$_2$:O$_2$ production is shown to be highly stable over the P3 and A3 N–TiO$_2$ catalysts (Fig. S7). In addition, no N$_2$ was observed in either experiment, which confirms that the nitrogen dopant is not being used as a stoichiometric hole scavenger. Furthermore, it is known that that N$^{3-}$ electro-oxidation to N$_2$ despite its high concentration still requires significantly higher applied potential than that of OH$^-$ to take place and the nitride oxidation can only be favourable in molten medium in the total absence of water[43]. Hence, it is shown that the hydrogen evolution is the product of photocatalytic water splitting, and that the N–TiO$_2$ is not consumed stoichiometrically with hydrogen yield.

The effect of polymorph and nitrogen-doping on activity will be discussed. From P0 being an inactive catalyst under visible light, the addition of nitrogen in P1, P2 and P3 leads to increasing photocatalytic activity. We attribute this activity directly to increasing concentrations of oxygen vacancies. It is widely known that nitrogen-doping in TiO$_2$ results in the formation energy for oxygen vacancies to decrease, as well as the decrease in band gap shown by the photo-absorption studies above. These oxygen vacancies result in lattice reduction and provide unpaired electrons to be trapped for eventual photo-excitation and proton reduction via excited Ti$^{3+}$ species[5]. Hence nitrogen-doping is known to increase the number of excitable charge carriers and provide a lower energy transition. Hence, it is predicted that there will be a correlation between the concentration of oxygen vacancies and photocatalytic hydrogen evolution. Actually, it has also been reported that the oxygen vacancies can interact with the surrounding water molecules, resulting in different adsorption and dissociation pathways[44,45].

In order to deconvolute the constituent phases of P25, the N-doped rutile and anatase catalysts were analysed by electron paramagnetic resonance (EPR), which is able to observe surface active sites through the characteristic unpaired electron signal at $g = 2.003$ of molecular oxygen from air occupying an oxygen vacancy as O$_2^-$ [46–48]. This signal is evidently present in both rutile and anatase materials indicating the presence of oxygen vacancies (Fig. 3). The signal was observed for the undoped R0 catalyst, which is attributed to the ease of formation of surface oxygen defects during synthesis. According to DFT calculations by Cheng and Selloni[11], 3.68 eV is required to create surface oxygen vacancies on rutile (110), which is more favourable than 4.15 eV of anatase (101). The signal increases considerably following initial nitrogen-doping yielding an oxygen vacancy

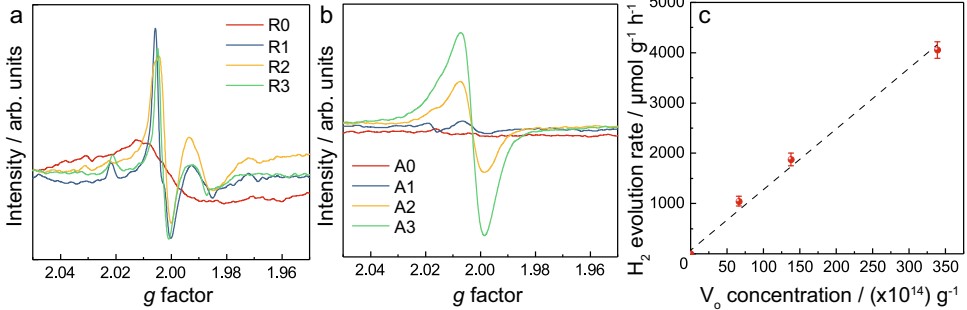

**Fig. 3 EPR spectra and their hydrogen evolution activity of rutile and anatase materials.** EPR spectra for (**a**) rutile and (**b**) anatase materials. **c** Hydrogen evolution of anatase catalysts under visible light against EPR-derived oxygen vacancy concentration. Error bars indicate the standard deviation.

| **Table 2 Oxygen vacancy concentrations derived from EPR spectra of rutile and anatase materials.** | | | |
|---|---|---|---|
| **Sample** | **Oxygen vacancy concentration/($10^{14}$) g$^{-1}$** | **Sample** | **Oxygen vacancy concentration/($10^{14}$) g$^{-1}$** |
| R0 | 1.36 | A0 | Not detected |
| R1 | 2.18 | A1 | 66.1 |
| R2 | 2.41 | A2 | 138 |
| R3 | 2.58 | A3 | 339 |

concentration of $2.18 \times 10^{14}$ g$^{-1}$ for R1. However, for higher nitrogen-composition rutile catalysts, the signal does not continue to increase significantly, with R3 not exceeding an oxygen-vacancy concentration of $2.6 \times 10^{14}$ g$^{-1}$ (Table 2). This indicates that the oxygen vacancies are mainly created on the local surface with no formation or migration to deeper layers, causing a limiting concentration. This again matches with the prohibitively high $V_o$ value of 5 eV in creating sub-surface oxygen vacancy in rutile[11]. As predicted, the low concentration of oxygen vacancies is correlated with the poor rate of hydrogen evolution for these catalysts (<1000 µmol g$^{-1}$ h$^{-1}$).

In contrast, the strength of the signal for anatase is clearly proportionate with the nitrogen content, indicating that the concentration of surface oxygen vacancies is governed by the concentration of dopant atoms. It is significant that the signal for anatase is over 100-times that of the highest rutile sample, reaching a concentration of $339 \times 10^{14}$ g$^{-1}$ for A3. At this concentration, the catalytic activity is far above any rutile catalyst, exhibiting the highest activity reported in this work (4000 µmol g$^{-1}$ h$^{-1}$). In fact, for anatase it is shown that a highly linear correlation exists between oxygen-vacancy concentration and photo catalytic activity (Fig. 3c). Hence, it is observed that oxygen-vacancy concentration increases with nitrogen-doping for both phases, with anatase exhibiting a much larger absolute concentration, which has a stronger non-limited dependence on nitrogen composition and a highly linear correlation with photocatalytic activity.

The mixture of anatase and rutile phases in P25 leads to an intermediate catalytic performance for P3; though interestingly, the activity of P1 and P2 are greater than A1 and A2, respectively. This improved performance is indicative of improved exciton separation by band-bending mechanisms which are often reported in rutile–anatase mixtures[49,50]. The higher work function of anatase is likely to result in downward bending and accommodation of electrons, facilitating hydrogen reduction. Inversely, the rutile accepts holes, facilitating oxygen evolution.

In summary, generally it has been shown that greater temperature ammonia treatment leads to higher nitrogen composition, which leads to lower formation energy and greater concentration of oxygen vacancies at a given temperature, which

leads to increased photocatalytic hydrogen evolution. The catalysts P3, A3 and R3 were also tested under combined UV/visible irradiation, with the absorption of visible light increasing the photocatalytic activity by approximately 50% (Fig. S5).

In the diffraction patterns for all nitrogen-doped anatase samples, anatase is the only phase present, which shows that these levels of nitrogen-doping do not cause major structural change. Despite the failure to detect bulk structural change by XRD, there are clearly indications that oxygen-vacancy formation is facilitated by nitrogen-doping at low nitrogen content. In order to detect changes in the sub-surface or bulk over this powder, we then studied the change in lattice parameters with respect to temperature by VT-SXPD.

**Anisotropic thermal expansion of anatase.** A discussion into the errors associated with this study is presented in the Supplementary Information (Tables S1 and S2, Fig. S7), including aspects of temperature-calibration, structure refinement, and data handling. Error bars are plotted on all graphs; the apparent absence of error bars is due to negligible errors.

Pristine and highly-doped P25 samples, P0 and P3, were analysed by high-resolution SXPD at various temperatures under atmospheric conditions. A Rietveld refinement was used to obtain the change in lattice parameter with respect to temperature. Fitting all patterns using a biphasic anatase-rutile model yielded a good fit, $R_{wp} < 5\%$, $GoF < 4$ (Table S3). To a certain extent, the lattice parameters exhibit typical thermal expansions reported previously in the literature[35,36]. The rutile phase indeed displays highly linear expansion, but interestingly the expansion of the anatase phase is distinctly non-linear (Fig. 4).

In order to see the distinctive temperature effect on the lattice parameters of anatase, the linear thermal expansion coefficients, $\alpha$, were calculated, in which $x$ is the lattice parameter and $T$ is the temperature (Eq. 1).

$$\alpha_x = \frac{1}{x}\left(\frac{\mathrm{d}x}{\mathrm{d}T}\right) \qquad (1)$$

Both $\alpha_a$ and $\alpha_c$ generally increase with increasing temperature, giving typical positive thermal expansion coefficients (Fig. 4b). Excitingly, centred around 200 °C, an unusual behaviour is clearly observed. At 150 °C, $\alpha_a$ starts to experience a dramatic decrease; meanwhile at the same temperature, $\alpha_c$ increases. Both changes peak at 200 °C, after which the trends gradually lessen until 350 °C after which the typical linear increase in thermal expansion is again observed. This behaviour occurs again when repeated with the same sample and with other samples. The similar onset and fading temperatures of the features indicate a common mechanism for both P0 and P3 but with the latter exhibiting greater deviations, which are confirmed to be real by the distinct separation of the values including their error. We attribute this

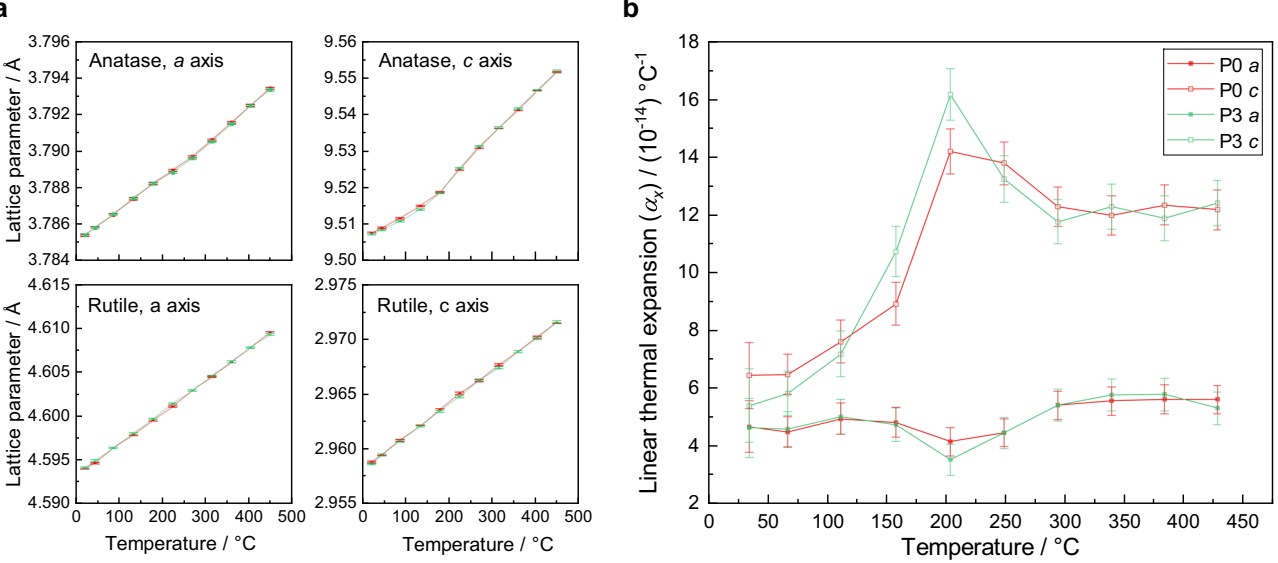

**Fig. 4 Thermal expansion of pristine and highly-doped P25 powders. a** Lattice parameters of anatase and rutile phases for P0 and P3 from room temperature to 500 °C. **b** Linear thermal expansion coefficient $\alpha$ for the *a*-axis and *c*-axis of the anatase phase in P0 and P3. Error bars indicate the standard deviation.

anisotropic linear thermal expansion to the structural reorganisation of anatase caused by oxygen vacancy formation.

As is well-known, in the anatase structure titanium is coordinated by six oxygen atoms coordinate sites as a Ti–O octahedron, for which tetragonal asymmetric is evident in the two distinct Ti–O bonds: four short equatorial bonds (*ab*-plane), and two slightly longer axial bonds (*c*-axis). This is in part due to inequivalent electron sharing between Ti–O, as the electron occupancy in $e_2$ orbitals undergoes Jahn–Teller like distortions. The thermal treatment of anatase induces greater vibration and weakening in the longer axial bond, causing a higher degree of increase in $\alpha_c$ than $\alpha_a$. This change in ligand environment will induce a contractive force on the equatorial bonds because the electron density of the titanium atom is sought to be maintained. As a result, the *ab* plane experiences an unusual decrease in thermal expansion coefficient over this temperature regime. Thus, this first mechanism accounts for the onset of anisotropic behaviour in anatase.

However, at higher temperatures of 200 °C, the inverse anisotropic behaviour is observed and the counter-distortion of which is caused by the formation of surface oxygen vacancies in the anatase phase. This is supported by the findings of our previous publication[34], in which we showed by variable-temperature EPR that oxygen vacancies begin to form at 200 °C (Fig. S8). Specifically, we believe that these are ordered sub-surface oxygen vacancies in P25 as measurable by VT-SXPD, which exist within the lattice under dynamic equilibrium with surface oxygen vacancies. The under-coordinated titanium sites that result from oxygen vacancies are namely two square pyramids with the vacancy in the *ab* plane, $Ti_{V,eq}$, and one with the vacancy in the *c*-axis, $Ti_{V,ax}$. The two electrons retained by the lattice following oxygen evolution ($O_o \rightarrow V_O^{2+} + \frac{1}{2}\ O + 2\ e^-$) are known to become localised in two separate Ti $3d$ orbitals forming two reduced $Ti^{3+}$ species[51]. We suggest that these three square pyramids in anatase structure, two of which are reduced ($d^1$), distort the bulk enough to be observed here at 200 °C. So above 200 °C, fully coordinated $TiO_6$ octahedra contribute by the first mechanism, while distorted and reduced polyhedra contribute a counter distortion by the second mechanism. These

balance above 350 °C where both thermal expansion coefficients return to a linear increase (Fig. 4b).

N–TiO₂ is also known to increase photocatalytic ability due to the increase in oxygen-vacancy concentration (see catalytic results at later section), which is achieved by a vastly decreased formation energy of oxygen vacancies[33]. As seen from the study of the temperature dependence of N–TiO₂ lattice parameters in P25 (Fig. 4b), there are small but significant differences between doped and non-doped samples. It is important to note that the anisotropic temperature effect of intrinsic (undoped) oxygen-vacancy formation is clearly smaller than that of extrinsic (dopant-facilitated) oxygen-vacancy formation. Hence, the difference between the curves yields the effect of extrinsic oxygen-vacancy formation from nitrogen doping. The magnitude of the anisotropy for the doped sample is anticipated to be larger than the non-doped material for both (*ab* and *c*) axes because in substitutionally doped anatase the Ti–N bond distance is larger than Ti–O[33]. For interstitial doping, the interstitial nitrogen $N^{3-}$ can also facilitate oxygen-vacancy formation and the accompanying reduction of $Ti^{4+}$ to $Ti^{3+}$, which are thought to be the active centres in photocatalysis. Also, the valence band of interstitial $N^{3-}$ lies above that of $O^{2-}$, allowing lower energy photoexcitation from diamagnetic nitrogen to Ti $3d$ orbitals[33], which is seen in the significant reduction in band gap discussed earlier. Thus, this observation is also consistent with the effects of higher degree of oxygen-vacancy formation to counteract the Jahn-Teller distortions at and above 200 °C.

We acknowledge that the presence of water could affect the adsorptive properties of surface oxygen vacancies in the samples, as well as the relative stability of the sub-surface vacancies[44,45]. Hence, the ex-situ powder diffraction may not be representative of the experimental conditions. However, we have previously reported that the EPR signals for N–TiO₂ materials increase in a similar way when in the presence of water vapour, liquid water, and in pure N₂, which suggests that the oxygen vacancies are still, at least regenerated at elevated temperatures under the photocatalytic water splitting conditions[34]. However, the exact nature of the water-adsorption effect is beyond the scope of this report so will not be discussed at great length.

**DFT study of oxygen vacancies in anatase**. In order to rationalise the anisotropic thermal properties of anatase, a small anatase supercell (space group: *P1*) was modelled and optimised by ab-initio DFT simulations (see DFT in SI). The supercell was orientated with the dominant (101) surface exposed to vacuum by terminating oxygen sites. The slab contained three layers of $TiO_2$ units labelled s (surface), ss (sub-surface), and b (bulk); the slab thickness was approximately 8.85 Å. In anatase (101) there are four oxygen environments and two titanium environments in each tri-layer stacked along the *c*-axis labelled $O\_1$, $O\_2$, $O\_3$ and $O\_4$ and two inequivalent titanium atoms labelled $Ti\_1$, $Ti\_2$ (Fig. 5). (Though relative to the vacancy, all sites are inequivalent.)

The calculated lattice parameters of pristine anatase structure matched with diffraction data taken account of the temperature factors ($a = b = 3.810$ Å, +0.63%; $c = 9.440$ Å, −0.68%). Oxygen-vacancy structures were optimised by removing atoms from $O_{s1}$, $O_{s2}$, $O_{s3}$ and $O_{ss3}$ (Table 3); the vacancy structures will be referred to as $VO_{s1}$, $VO_{s2}$, $VO_{s3}$ and $VO_{ss3}$, respectively.

It is interesting to note from the optimised slab energies that the lowest energy vacancy-structure is that of the ordered sub-surface vacancy ($VO_{ss3}$) rather those with surface vacancies ($VO_{s1}$, $VO_{s2}$, $VO_{s3}$). The difference in stability between the two lowest energy structures ($VO_{s3}$ and $VO_{ss3}$) is 0.37 eV. Anatase (101) has a relatively low surface energy compared to other facets, which requires more energy to create surface vacancies. Consequently, in contrast to other $TiO_2$ facets, as well as other oxides, we and others have shown that sub-surface oxygen-vacancy formation is more energetically favourable. This was also demonstrated by Cheng and Selloni[12] who reported a stability difference between these sites of 0.46 eV, corresponding to a relative probability of a surface vacancy of $4.0 \times 10^{-9}$ and $1.6 \times 10^{-3}$ at 300 K and 900 K, respectively. Their results support our simulations that the lowest energy vacancy-structure is yielded by removing the $O_{ss3}$ sub-surface oxygen than surface oxygen. In addition, they demonstrated that the next-deeper site of $O_{b3}$ is similarly stable to $O_{ss3}$, and in contrast that all sub-surface and bulk vacancies in rutile are highly disfavoured. It is the stable sub-surface/bulk oxygen vacancies that are believed to be crucial in the superior photoactivity of anatase (101), as it allows the mobility of vacancies away from the surface, delimiting the total concentration of vacancies. Indeed, according to the calculations by Fiscaro et al.[52], oxygen vacancies can travel along the *P1* *c*-axis from a higher energy $O_{s1}$ vacancy to the global-minimum vacancy at $O_{ss3}$. Elsewhere, the surface to sub-surface ($O_{s1} \rightarrow O_{ss3}$) oxygen-diffusion activation energy is calculated to be 67 kJ mol$^{-1}$ [11], which matches well with our EPR-derived activation energy of 63 kJ mol$^{-1}$ (Fig. S9).

We acknowledge that our calculations use the conventional PBE functional[53] with a dispersion correction scheme[54], which lack some of the computationally intensive and complex recent improvements in the modelling of $TiO_2$[55]. With respect to hybrid potentials, in a study by Li et al.[56], they showed that a PBE functional and a hybrid functional yielded similar results and identical conclusions. As for the Hubbard factor, it has been reported that standard DFT can adequately predict structural and optical properties in a range of $TiO_2$ materials[57] and specifically for the anatase (101) surface[58]. In addition, elsewhere it has been conceded that a universal value for $U$ cannot reproduce $TiO_2$ properties, often leading to worsening of the accuracy of the crystal structure[59]. The energies of the various optimised structures and sites reported in this work agree with previous reports, and support our observations by EPR, SXPD and photocatalytic activity, which have been discussed above. Hence, our simple and computationally efficient calculations can be considered sufficient for this system.

The structural effects of the $O_{ss3}$ vacancy will now be interrogated, as we show here significant reorganisation occurs in the surrounding atoms in order to stabilise the vacancy. Interestingly, the Ti–Ti distances in the *ab*-plane are elongated (+0.714 Å) causing *ab*-expansion; meanwhile $TiO_2$ along with *c*-axis is contracted (−0.152 Å) causing *c*-contraction. It is these changes that we also observed in the anisotropic thermal expansion activity due to oxygen vacancy formation above 200 °C by the SXPD. The low energy for the sub-surface oxygen-vacancy formation can facilitate oxygen-vacancy formation and mobility, as indicated by EPR (Fig. 3), resulting in highly efficient surface photocatalytic activity. These distortions will then counteract the *c*-axis thermal expansion in bulk with increasing temperature. We also note that the low energy sub-surface $VO_{ss3}$ structure yields a highly distorted $O1$–$TiO_2$ moiety ($\Delta\theta = +72.7°$) (Table 3). Further explanation of this structural distortion is discussed in Fig. S10.

Crystallographic techniques are generally unable to detect disordered surface oxygen vacancies; however, the surface oxygen vacancies in nanocrystalline $TiO_2$ anatase (101) structure are

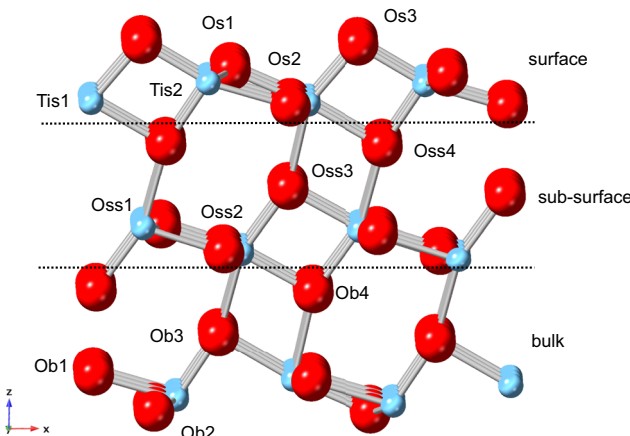

**Fig. 5 The model of titanium oxide nanoparticle showing different oxygen species on surface, subsurface and bulk.** Pristine anatase (101) supercell showing assigned depth environments and inequivalent (relative to surface, not vacancy) oxygen (red) and titanium (blue) sites in each. Depth: s (surface); ss (sub-surface); b (bulk). Volume of vacuum not shown. Note, there is no $O_{s4}$ due to surface termination.

**Table 3 DFT results showing the absolute energy, relative energy to minimum structure VO$_{ss3}$, and changes in local bond distances and bond angles relative to the pristine anatase (101) slab.**

| Structure | Energy/eV | $\Delta E$ relative to $VO_{ss3}$/eV | $\Delta$ Ti-Ti / Å | $\Delta$ Ti-O1/ Å | $\Delta$ Ti-O2/Å | $\Delta\theta_{O1-Ti-O2}$/° |
|---|---|---|---|---|---|---|
| $VO_{s1}$ | −937.38 | 1.04 | 0.57 | N/A | 0.02 | 36.3 |
| $VO_{s2}$ | −937.64 | 0.78 | 0.62 | −0.04 | N/A | −24.4 |
| $VO_{s3}$ | −938.05 | 0.37 | 0.34 | −0.15 | 0.10 | −5.1 |
| $VO_{ss3}$ | −938.42 | 0.00 | 0.71 | −0.08 | −0.15 | 72.7 |

clearly unstable with respect to sub-surface ordered oxygen vacancies. Hence significant oxygen mobility from sub-surface sites to surface vacancy sites under dynamic equilibrium is anticipated. This accounts for subtle structural change in the sub-surface lattice which has been observed by VT-SXPD as anisotropic thermal expansion of the anatase phase in P25. This can be directly correlated to the oxygen-vacancy concentration of the anatase phase, nitrogen doping and temperature.

This combined study by DFT and VT-SXPD has revealed the significant 'storage' of sub-surface vacancies formed by fast oxygen transfer between surface sites in anatase at elevated temperature, which are important for interaction with excitons during photo-excitation in the whole structure. We suggest that the large sub-surface oxygen-vacancy storage and facile oxygen mobility clearly indicates that the anatase phase is a superior photocatalyst for water splitting to $H_2$ and $O_2$, especially compared with rutile which only has limited and immobile surface vacancies[12]. However, catalysis is highly facet-sensitive and site-sensitive and is unlikely the same for both facile reduction and oxidation reactions. It is still not conclusively known it is not yet clear whether surface or lattice oxygen vacancies (sub-surface) play the key role in catalysis over N–TiO$_2$. It is important to note that the anisotropic thermal expansion was observed in both pure TiO$_2$ and N–TiO$_2$. These computational results also indicate that it is favourable for sub-surface oxygen vacancy formation in pure TiO$_2$. But, we have also shown that the nitrogen-doping of N–TiO$_2$, either as an interstitial or substitutional dopant, can narrow the band gap and increase the concentration of the oxygen vacancies, resulting in increased photocatalytic water splitting by visible light. This can demonstrate the unique roles of N doping in the N–TiO$_2$ for the photocatalytic water splitting at elevated temperature.

Here the in-depth structural study of rutile, anatase and mixed phase P25, with and without nitrogen-doping, by variable-temperature synchrotron X-ray powder diffraction (VT-SXPD) indicates that there exists a direct correlation between the anisotropic thermal expansion in lattice parameters and activity with respect to temperature.

**A new titanium oxynitride structure.** High-resolution SXPD of the pure anatase and pure rutile samples was carried out at BL02B2 at SPring-8, Japan. The patterns for the rutile samples that were nitrogen-doped at different temperatures did not significant differences, only small non-systematic variation in lattice parameter and increasing particle size with increased temperature conditions due to microcrystalline sintering (Fig. S11). Thus, the rutile phase in P25 is a minor or spectator phase during nitrogen-doping. Apart from the low level of nitrogen-doping in anatase (N–TiO$_2$), by adapting the same doping procedure, we show that it was possible to derive the structure in a high level of nitrogen doping at high temperature ammonia treatments. On the other hand, pure anatase and P25 materials exhibited additional peaks that are consistent with a cubic $Fm\bar{3}m$ rock-salt phase. As discussed earlier in this report, XPS suggested the presence of both interstitial and substitutional nitrogen in this sample (Fig. 1). These reflections grew in intensity as the doping temperature was increased for pure anatase and P25 materials, hence are correlated with nitrogen composition. The phase strongly resembled titanium nitride Ti$^{3+}$N$^{3-}$ which is formed by nitridation of anatase accompanied by oxygen evolution. For very high temperature nitridation of anatase at 750 °C (A4), the peaks become more dominant and narrower due to microcrystalline sintering (Fig. 6). However, the A4 TiN reflections were shifted to low angle in comparison to the low nitrogen-doped materials, indicating an expanded unit cell. Of additional note, the presence of rutile was

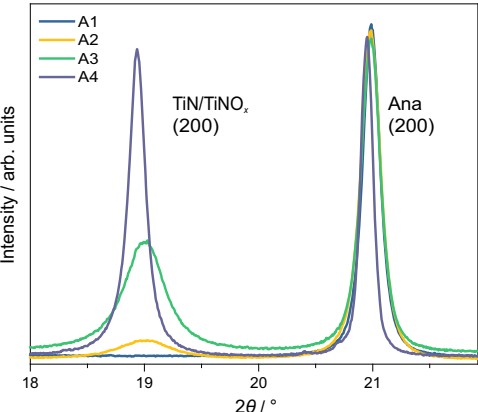

**Fig. 6 SXRD patterns of anatase with different N contents.** SXPD patterns of anatase samples A0–A3, as well as a high-temperature sample A4 prepared at 750 °C. Intensities normalised due to different data-collection parameters.

not observed in the diffraction patterns, despite the prolonged thermal treatment at 750 °C being sufficient for the conversion of anatase into rutile which relative to anatase is the more thermodynamically stable polymorph, as previously seen in XRD studies[30].

This assumption was applied to a biphasic Rietveld refinement of A4 comprising a tetragonal $I4_1/amd$ anatase phase and a cubic $Fm\bar{3}m$ titanium nitride phase. The model yielded chemically reasonable lattice parameters for anatase ($a_{ana}$ = 3.7852 Å, $c_{ana}$ = 9.5106 Å); though the value for 'titanium nitride' phase ($a_{TiN}$ = 4.1843 Å) was actually lower than the known lattice parameter in pure phase (4.235 Å[60]). This unit cell is notably more similar to that of $Fm\bar{3}m$ titanium(II) oxide (4.184 Å[61]) which is formed at high temperatures from titanium(IV) oxide and titanium metal. Overall the fitting was reasonable ($R_{wp}$ = 3.38%, $GoF$ = 1.98) (Table S3). However, though the Ti$^{3+}$ occupancy refined to near unity (0.9998), the occupancy of the nitrogen atom refined to a chemically unreasonable value of 1.48 (see Fourier difference electron density map for unity occupancy in Fig. S12). This is due to nitrogen being in the N$^{3-}$ anion in the structure rather than the neutral atom used (10 electrons rather than 7) since the fitting depends critically on both electron density of atom and site occupancy. As the atomic scattering factors for N$^{3-}$ are unavailable in the refinement programme, the isoelectronic O$^{2-}$ was used instead (slightly differing electron density). The occupancy of this atom then refined to a more chemically sensible value of 1.01 ($R_{wp}$ = 3.39%, $GoF$ = 1.99).

The phase information derived from the Rietveld refinement was applied to the observed structure factors in order to synthesise electron density maps (Fig. 7a). The rock-salt structure of highly electron dense titanium Ti$^{3+}$ and lower-density nitrogen N$^{3-}$ is expected. However, unexpected electron density is also observed at the tetrahedral hole positions (0.25, 0.25, 0.25) (Wyckoff: 8c). This is more clearly observed in the Fourier difference electron density map, shown in 2D and 3D (Fig. 7b, c). Introducing a neutral oxygen atom into the tetrahedral hole further improves the fit ($R_{wp}$ = 3.35%, $GoF$ = 1.96) (d) and the Fourier difference map becomes more homogeneous (Fig. S13). The resulting fluorite structure is displayed in Fig. 8e. The occupancy of the site is 0.03 resulting in the formula Ti$^{3+}_{0.95}$N$^{3-}_{1.07}$O$_{0.03}$, which has a net charge of −0.38 $e$. Refinements yield slightly better fits when titanium atoms are modelled as tetravalent Ti$^{4+}$ rather than trivalent Ti$^{3+}$ showing a degree of charge transfer from Ti to O when O is interstitially

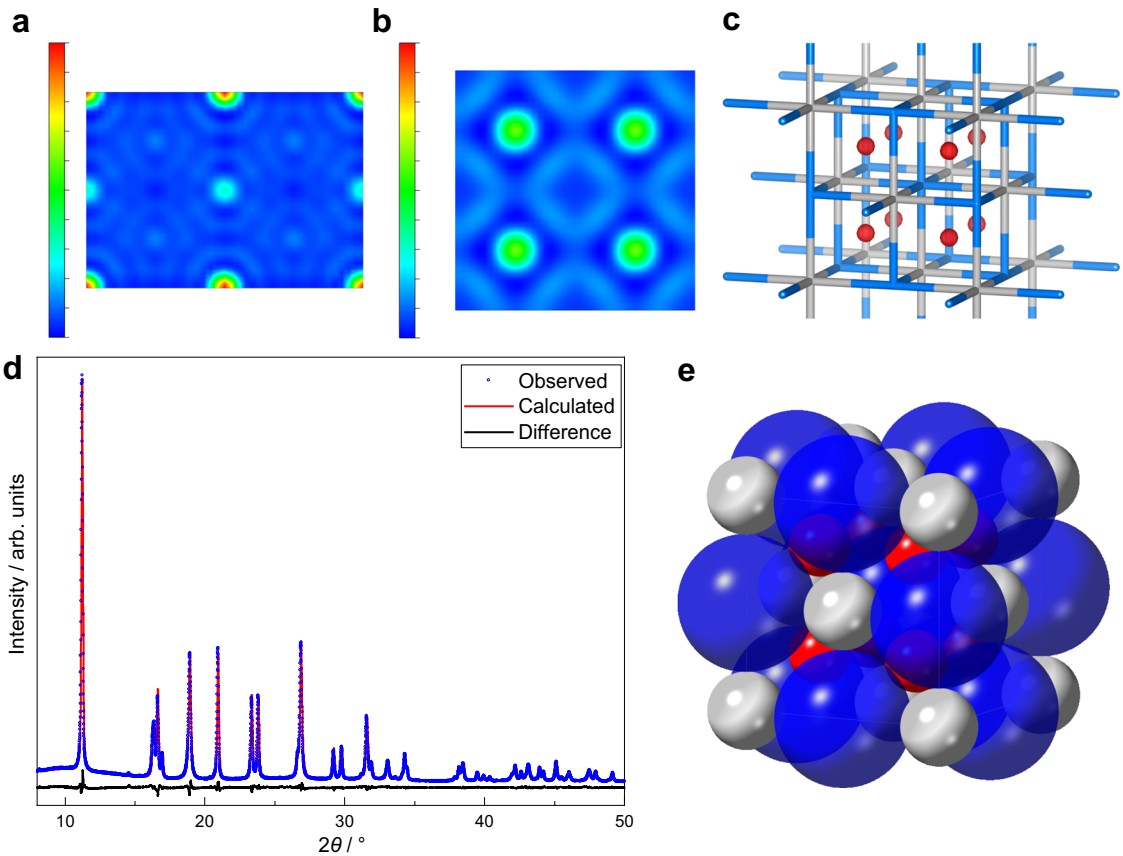

**Fig. 7 Phase identification of TiNO$_x$ by Rietveld refinement.** Electron density maps of the TiNO$_x$ unit cell: 2D cross-sections of electron density derived from the Fourier transform of (**a**) $F_{obs}$ at one unit cell length $a$ from the origin along (011); (**b**) $F_{diff}$ (=$F_{obs} - F_{calc}$) at 0.25$a$ along (001); (**c**) iso-surface of electron density at 0.1 (red); grey: titanium, blue: nitrogen. **d** Fitting of Rietveld refinement of A4 at room temperature consisting of an anatase phase and a titanium oxynitride phase: nitrogen atom is modelled as O$^{2-}$ and interstitial oxygen anion modelled as O atom at tetrahedral hole. **e** Corresponding structure solution for TiNO$_x$: blue: titanium, red: nitrogen, white: oxygen.

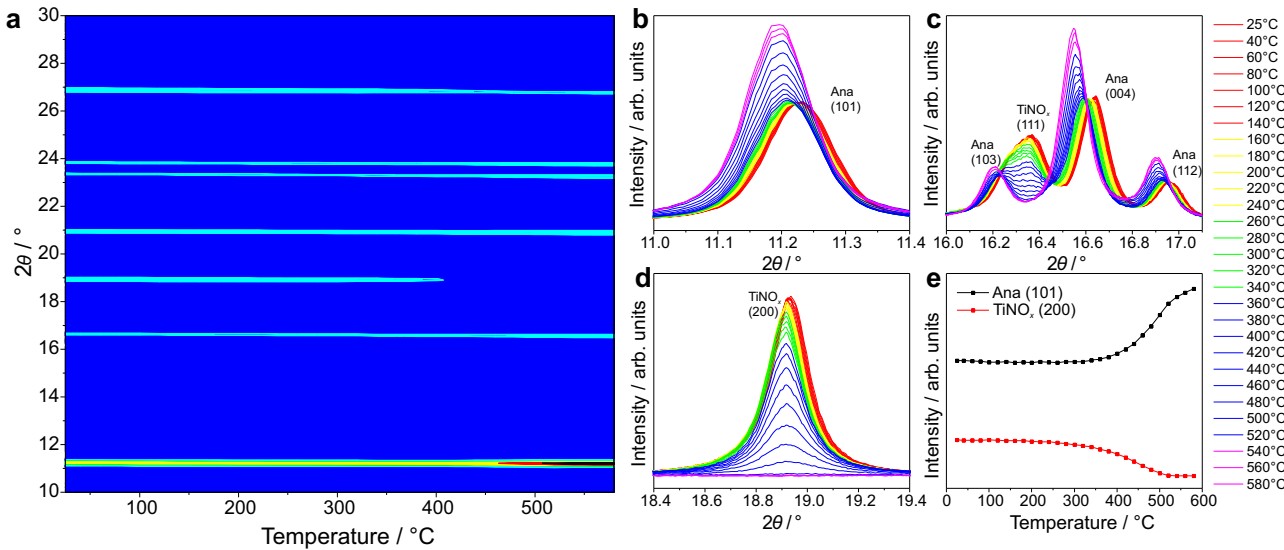

**Fig. 8 Crystallographic variable temperature analysis of A4. a** In-situ VT-SXPD patterns; (**b**–**d**) enlarged regions showing the decomposition of TiNO$_x$ and increase in anatase intensity; (**e**) intensity of anatase (101) and TiN (200) plotted against temperature (negligible standard deviation not plotted).

added. It is therefore conceivable that significant N$^{3-}$ inclusion in the previously discussed N–TiO$_2$ anatase phase by high temperature ammonia nitridation generates significant concentrations of Ti$^{3+}$ and V$_O^{2+}$, which can create the new nitride 'Ti$^{3+}$N$^{3-}$' phase with residual O to occupy the interstitial sites. Although the main phase corresponds to anatase for the former and nitride for the latter, both can thus be regarded as oxynitride structures.

VT-SXPD data was also collected on beamline BL02B2, SPring-8 for the thermal oxidation of A4 in air from room temperature to 580 °C. The sample was composed of 67.8%wt anatase (N-TiO$_2$) and 32.2%wt TiNO$_x$ phases. The TiNO$_x$ is stable until 240 °C, after which the peaks of this structure decrease in intensity until the complete disappearance of any reflections by 520 °C (Fig. 8) (refined values for powder composition shown in Fig. S14a). Anatase is shown to be the dominant decomposition product of TiNO$_x$ phase in air by losing nitrogen and gaining oxygen, shown by the highly correlated increase in the intensity of the anatase phase (likely the reverse of the process that formed this phase, i.e., nitridation of anatase by ammonia). Indeed, by the same refinement it is shown that the occupancy of interstitial oxygen from refinement is indeed gradually increasing before the phase conversion (Fig. S14c). Thus, this clearly shows that the interchangeable phase between anatase and nitride is dependent on nitrogen and oxygen availability in the oxynitride structures at elevated temperature.

According to the catalysis study reported in the next section, the modified oxynitride structure of TiN can also contribute to the high photocatalytic activity under visible light of highly nitrogen-doped composite TiO$_2$ materials.

**Temperature dependence of catalytic activity.** All catalytic data up to this point has been carried out at 270 °C. Here, the temperature dependence of photocatalytic activity of all nitrogen-doped catalysts were studied at different temperatures. The effect of temperature was first investigated over samples of highest nitrogen-doping, A3, R3 and P3. For A3 and P3 there is a clear onset of photocatalytic activity at 200 °C with exponential increase in the rate under visible light with temperature (Fig. 9). Meanwhile, the activities of R3, although slightly increased at elevated temperatures, remained at a low level. This onset in activity coincides with the counter Jahn–Teller mechanism shown in Fig. 4b attributed to the onset of oxygen-vacancy formation in the asymmetric nitrogen-doped anatase phase at 200 °C. Indeed, the activation barrier for hydrogen evolution has been calculated to be 60.6 kJ mol$^{-1}$, which is in good agreement with the activation barrier for oxygen-vacancy formation of 62.5 kJ mol$^{-1}$, which we derived from variable-temperature EPR (Figs. S8 and S9). In addition this formation energy is in good agreement with the reported activation energy for oxygen-vacancy mobility in anatase of 67.5 kJ mol$^{-1}$ [11], which indicates that the formation of

oxygen vacancies in anatase is limited by the diffusion of vacancies into the sub-surface.

It is clear that the photocatalytic performance is clearly found to be strongly dependent on the nitrogen-doping levels and sub-surface oxygen-vacancy concentration, since the photocatalytic activities of the nitrogen-doped P25 and anatase samples significantly increase with greater nitrogen-doping; meanwhile all nitridation temperatures for nitrogen-doped rutile exhibited similar activities due to similar oxygen-vacancy concentrations as shown in the EPR results.

At low N-doping, the interstitial/substitutional N$^{3-}$ in N–TiO$_2$ increases the concentration of V$_O^{2+}$ and Ti$^{3+}$ and also reduces the band gap to the visible light regime, as previously demonstrated. Although it is beyond the scope of this study how one could feasibly control the ratio of interstitial and substitutional doping, Lynch[37] and Zhang[38] are amongst the researchers to report fine control. Nevertheless, the formation of H$_2$ is attributed to H$^+$ reduction of water on surface excited electrons trapped at surface Ti$^{3+}$, and O$_2$ formation attributed to oxidation of hydroxyl ions (OH$^-$) from water over surface V$_O^{2+}$ during photolysis, which are in equilibrium with the sub-surface/bulk defects in such oxynitride structure. Similarly, at high N loading, OH$^-$ from water over unfilled interstitial sites in TiNO$_x$ phase can be oxidised to O$_2$ and provide electrons to regenerate Ti$^{4+}$ to Ti$^{3+}$ after H$_2$ formation from H$^+$ reaction allowing the same pathway for photoexcitation in band gap modification in the oxynitride structures. The dissociated H$^+$ and OH$^-$ ions that are adsorbed on the surface, the concentrations of which are dependent on pH, can generate a local electric field, which prolongs the lifetime of surface excitons, as we have confirmed by time-resolved photoluminescence spectroscopy (Fig. S15). Therefore, the enhanced photocatalytic performances are attributed to the suppressed electron-hole recombination process, which we have discussed in more detail elsewhere[34]. We also postulate that the anatase-oxynitride mixture may benefit from band-bending effects, which result in stronger charge separation, greater exciton lifetime and enhanced catalytic activity, as has been shown for other anatase systems[49,50]. Like rutile, the work function of titanium nitride is lower than anatase. This indicates that the titanium-nitride-like oxynitride phase will confer similar charge separation as discussed for P1 and P2 samples earlier here.

The use of elevated temperature and nitrogen-doping have been empirically shown to increase TiO$_2$ photocatalytic activity, with the latter broadening the absorption profile from UV to the near-IR region[51]. In a recent investigation[34] and review[62] of ours, the use of elevated temperature was reported to facilitate the oxygen-vacancy regeneration of nitrogen-doped TiO$_2$ photocatalysts, and enhance the photo-generated charge separation via surface polarisation, both of which contribute to improved kinetics. It was also shown that elevated temperature is a more effective method of increasing H$^+$ concentration than varying the pH of the water. Though elevated temperatures may seem impractical and cost-inefficient, we reported a complete and efficient visible-light-driven water splitting system, in which only light is required to reach the high temperature by using an optical floating-zone light furnace. This method and other light concentrator configurations, primarily compound parabolic cylinder reflectors[63], can provide enhanced light irradiation and temperature for small/medium size applications, which include photo-voltaic cells[64], photomicroreactors[65], photocatalytic waste degradation[66–68], and photocatalytic hydrogen evolution by TiO$_2$ materials[34,69], with fully sunlight dependent hydrogen evolution photocatalysts reported[70].

With respect to the photocatalytic configuration, other preliminary study are being carried out to further improve the efficiency of this process. The possibility of using water vapour

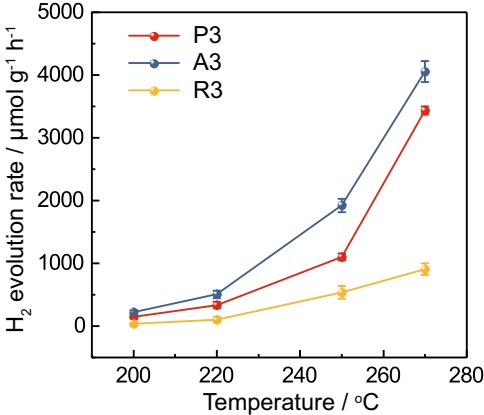

**Fig. 9 Photocatalytic water splitting activity study of the N-doped TiO$_2$ samples.** Photocatalytic water-splitting activities over P3, A3 and R3 photocatalysts at different temperatures under visible-light illumination (red: P25; blue: anatase; yellow: rutile). Error bars indicate the standard deviation.

instead of liquid water, which possesses lower heat capacity and yields lower pressures at a given temperature, is highly promising. In addition it makes the reaction more practical, controllable and easier to operate. Further studies on this subject will be carried out.

In summary, an unusual anisotropic thermal expansion of anatase has been observed at 150 °C, namely that the *c*-axis exhibits increased thermal expansion while that of the *a*-axis decreases. This has been attributed to the increased weakening of the long Ti–O$_{ax}$ bond. At 200 °C the phenomenon is countered which is attributed to the Jahn–Teller contraction of 5-fold coordinated Ti$^{3+}$ polyhedra formed by oxygen vacancies. The orthogonal distortions appears to balance by 350 °C. For low levels of nitrogen-doping in N–TiO$_2$ anatase the inclusion of nitrogen atoms, either as an interstitial or substitutional dopant, can increase the concentration of oxygen vacancies and Ti$^{3+}$, and narrow the band gap resulting in photocatalytic water splitting by visible light. At high N-doping, a new phase of cubic titanium nitride with interstitial oxygen at high temperature ammonia decomposition on anatase is for the first time identified. Both structures can be described as oxynitrides, are crystallographic related, and share the same photocatalytic mechanism. They are active and kinetically stable in photocatalysis at elevated temperatures of below 300 °C, in comparison to the photocatalytically inert rutile.

## Methods

**Materials**. Analytical grade fine powder titanium dioxide (Degussa P25, 75% anatase, 25% rutile); Anatase TiO$_2$ (IshiharaSangyo, Japan); Rutile TiO$_2$ (99.995%, Sigma-Aldrich); Ammonia gas (anhydrous, BOC) were used without further purification.

**Synthesis of N-doped TiO$_2$**. Phase pure anatase TiO$_2$ powder (denoted as A0) was nitrogen-doped using an ammonia treatment by annealing under ammonia flow at temperatures between 550 °C and 620 °C for 8 h. The degree of nitrogen doping was controlled by the temperature of the reaction which was set at 550 °C, 600 °C and 620 °C yielding samples denoted as A1, A2 and A3, respectively. Likewise, the ammonia treatment at different temperatures was also performed on rutile TiO$_2$ (R0) and P25 TiO$_2$ (P0) powder samples. R0 was treated with ammonia flow at 650 °C, 700 °C and 750 °C, which are denoted as R1, R2 and R3, while P0 at 550 °C, 600 °C and 620 °C, denoted as P1, P2 and P3.

**Photocatalytic water-splitting activity tests**. The photocatalytic performances were evaluated by measuring the amount of hydrogen and oxygen evolved from the water splitting reaction, which were carried out in a closed 25 mL stainless-steel autoclave equipped with two quartz windows (10 mm in diameter, 18 mm in thickness). Typically, 5 mg of photocatalyst was added to 10 mL of Milli-Q H$_2$O under vigorous magnetic stirring (750 rpm); after being sealed the system was pressurised with 2 bar of Ar gas. The reactor was heated to the designated temperature, after which a tungsten light (70 W, Glamox Professional 2000) was applied through the quartz windows to provide visible-light irradiation. After 2 h of reaction, the autoclave was cooled naturally to room temperature. The amounts of hydrogen and oxygen were measured by gas chromatography (GC) equipped with thermoconductivity detectors (TCD).

**Thermal expansion of anatase, N-anatase, P25 and N-P25**. Synchrotron X-ray powder diffraction (SXPD) data was collected at the High Resolution Powder Diffraction beamline I11 at Diamond Light Source, UK (DLS)[71]. The energy was specified as 15 keV, with the wavelength calculated to be 0.824875(1) Å (~15.031 keV) and the angular zero-error to be +0.004675° in 2θ by using a NIST silicon standard. Samples of P25 and N-P25 were loaded into borosilicate capillaries of diameter 0.5 mm, and analysed by Debye-Sherrer geometry using a multi-analyser crystal (MAC) diffractometer over an angular range of 2–150° in 2θ over a 30-min exposure period. Scans were conducted at room temperature and 50 °C intervals between 100 °C and 500 °C. Heating was conducted using a hot-air blower at 10 °C min$^{-1}$, with an equilibration time of 2 min allowed before each scan. The calibration of the sample temperature was conducted prior to investigation by using the well-known thermal expansion of platinum[72] (Tables S1 and S2, Fig. S1). The experiment was repeated using N-P25 (P3).

**Phase analysis of high temperature synthesis**. SXPD data of pristine and nitrogen-doped anatase, rutile and P25 were also collected at beamline BL02B2 at SPring-8, Japan[73]. Samples were loaded into borosilicate capillaries of diameter 0.5 mm, and analysed by Debye-Sherrer geometry using a position-sensitive MYTHEN detector over an angular range of 4–78° in 2θ over a 10-min exposure period (sum of 2 × 5 min). The beam energy was shown to be 17.98029(9) keV by refinement of a ceria standard, with an angular zero-error of −0.0015°. Each sample was scanned at room temperature and at high temperature 500 °C.

A high degree of nitrogen-doping was achieved for anatase (A4) by increasing the nitrogen temperature to 750 °C from the previous highest temperature of 650 °C. The sample was scanned over an exposure time of 2 × 100 s at room temperature and between 60 °C and 580 °C at 20 °C intervals. All structural refinements were carried out in TOPAS[74], and VESTA[75] was used for electron-density visualisation.

**DFT calculations**. Spin-polarised density function theory (DFT) calculations were performed by using the Vienna Ab-initio Simulation Package (VASP) programme[76,77] within the projector-augmented wave (PAW)[78] to explore geometries and electronic properties of the (101) facet of TiO$_2$ anatase. The exchange-correlation interactions were described with the generalised gradient approximation (GGA)[79] in the form of the Perdew, Burke and Ernzernhof (PBE) functional[53]. The kinetic energy cut-off for the plane-wave basis set was set as 450 eV, and the distance of vacuum layer was set to be more than 15 Å, which is sufficiently large to avoid interlayer interactions. The Grimme van-der-Waals/dispersion-correction scheme DFT-D3[54] was applied on the anatase TiO$_2$ (101) surface. The pristine slab contained three TiO$_2$ layers, totalling 108 atoms: 36 Ti and 72 O atoms. The electronic self-consistent field (SCF) tolerance was set to 10$^{-5}$ eV. Fully relaxed geometries and lattice parameters were obtained by optimising all atomic positions until the Hellmann–Feynman forces were less than 0.02 eV Å$^{-1}$. The *k*-points samplings with a gamma-centred Monkhorst–Pack scheme[80] were 8 × 8 × 1 for structural optimisations.

## Data availability

The authors declare that the data supporting the findings of this study are available within the paper and its supplementary information files. All relevant data are available from the authors.

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

## Acknowledgements

The support for this project from the EPSRC in the United Kingdom (Grant EP/K040375/1) is gratefully acknowledged. We thank Qingming Deng of Huaiyin Normal University, China for the DFT calculations.

## Author contributions

C.F. prepared and characterised all Ti containing catalysts by XPS, EPR, and VT-SXPD and refinements. Y.L. performed photocatalytic experiments and analysed TRPL data. T.C. characterised the samples by electron microscopy. S.D. and C.T. assisted in VT-SXPD collected at both Diamond Light Source, UK and SPring-8, Japan. C.F., Y.L. and S.C.E.T. wrote the paper in discussion with S.D. and C.T. S.C.E.T. supervised the overall project.

## Competing interests

The authors declare no competing interests.
