## [Peer Review File · Nature Communications]

REVIEWER COMMENTS

Reviewer #1 (Remarks to the Author):

The authors have written here a valuable paper that presents a comprehensive examination of crystallography and defect structures in anatase and rutile polymorphs, and importantly considers the role of nitrogen doping which. The field is awash with anecdotal observations into the roles of phase mixtures and nitrogen doping in titania photocatalysts. The mechanistic insights of this work will help researchers better explain behaviour observed in titania materials (including nitrated systems) and will hopefully contribute towards the progression of applied titanium dioxide materials in water-splitting. The energetically unfavourable surface vacancy formation in rutile relative to anatase is made clear here. The role of Nitrogen doping in modifying the surface oxygen vacancy structures in anatase is examined with a focus on the significance in water splitting applications. This offers valuable information towards rational materials design.

The work is certainly worthy of dissemination and will be of value to numerous other researchers in the field of titania photocatalysts.

I suggest several minor changes to raise the value of the information communicated in the manuscript:

1. Much of the conclusions in the manuscript hinge upon the accuracy of lattice structure determined by Rietveld Refinement. In particular with biphasic mixtures, Rietveld refinements have a certain margin of error. Authors should make efforts to ascertain a possible margin of error regarding anisotropic lattice expansion found here and critically examine potential sources of experimental or methodological error in this measurement.
2. Authors identify interstitial and substitutional nitrogen doping. If possible, this should be quantified, with some information regarding how one could feasibly control the ration of interstitial and substitutional doping in an optimal way.
3. Some insights into the improvement of water splitting rate through materials design should be given
4. An explanation of whether enhanced performance is observed in biphasic mixtures would be welcome and if this can be mechanistically explained. In anatase rutile mixtures, improved performance is often reported to arise due to improved exciton separation by band bending mechanisms. Is a similar mechanisms predicted to occur in biphasic anatase / oxynitride phase assemblages?
5. The role of oxygen vacancies on surface adsorption needs to be considered. The nature of water adsorption, molecular or dissociative is expected to be influenced by oxygen vacancies in doped and undoped materials. It is also possible that water adsorption stabilizes surface oxygen vacancies as opposed to sub-surface ones.

Reviewer #3 (Remarks to the Author):

In this study, the authors carried out a structural study of rutile, anatase, mixed phase TiO₂ and their corresponding N-doped samples. The effect of calcination temperature on the doping impurities, structural defects, vacancy concentration were investigated using VT-SXPD, DFT, etc. Anisotropic thermal expansion in lattice parameters was discovered, and the authors claimed the clarification of relationship between sub-surface oxygen vacancy, nitrogen-doping level and photocatalytic activity. Although the work describes interesting findings, I am not very confident that the findings are

important enough to be of general interests to Nature Comm. readership.

N-doped TiO₂ has been extensively studied but this study failed to demonstrate any significant improvements of N-doped TiO₂ for overall water splitting. The authors carried the water splitting at elevated temperature and pressure condition in an autoclave photoreactor. Such photocatalytic reactivities observed under extreme conditions have little practical meaning. In addition, the importance of the experimental findings is not well justified for the following reasons.

1. There is no evidence of overall water splitting. Is there a stoichiometric evolution of O₂ gas? What about the possibility of N₂ evolution as a result of the N anion oxidation? Are N anions acting as a hole scavenger for H₂ evolution? Prove that this reaction is photocatalytic by carrying out the multi-cycle photoreactions in the same batch of photocatalyst. Show if N₂ gas is evolved or not during the evolution of H₂.
2. The photocatalytic activity of N-TiO₂ for water splitting seems to be negligible at ambient temperature. What is the use of the water splitting under extreme conditions of high temperature and pressure? If you find some important information from this high temperature study, such knowledge should be demonstrated to improve the photocatalytic activity of N-TiO₂ under ambient condition, which is completely absent in this study.
3. I do not see any clear reasoning how unusual anisotropic thermal expansion in lattice parameters is related with the photocatalytic activity. Such enhanced activity at higher temperature might be generally observed in other semiconductor photocatalysts as well. As shown in Figure S1, the pure TiO₂ and N-TiO₂ show the similar temperature dependence qualitatively. Is there any evidence that this phenomenon is unique to N-TiO₂ only?

After further addressing the above points, this study should be better suited to a more specialized journal (material, catalysis, or physical chemistry).

Point-to-Point Reply to Reviewers' Comments

We sincerely thank all the reviewers for their time and valuable comments on the manuscript. These feedbacks are indeed very helpful for us to improve the presentation of this work. Both Reviewer 1 and 2 clearly recognised the novelty and showed the excellent understanding of our work. In light their construction comments, we have performed some new experiments (new data EPR, stability test), data retreatments (quantification for interstitial and substitutional nitrogen doping), reinterpretation (modelling and nature of oxygen vacancy) and revised discussions in the revisions. Reviewer 3 however, showed some misunderstanding and concerns of our high temperature photocatalysis, presumably due to the unclearness of our original manuscript without providing sufficient background information for him/her. In this revision, we have abstracted some previous results in the SI with given references and rewritten both the discussion and conclusion. Notice that we have provided the answers to all the reviewers' questions point by point and corrected the main text and supplementary information (SI) accordingly. Changes are marked in yellow in the revised manuscript and SI.

Reviewer #1 (Remarks to the Author):

The authors have written here a valuable paper that presents a comprehensive examination of crystallography and defect structures in anatase and rutile polymorphs, and importantly considers the role of nitrogen doping which. The field is awash with anecdotal observations into the roles of phase mixtures and nitrogen doping in titania photocatalysts. The mechanistic insights of this work will help researchers better explain behaviour observed in titania materials (including nitride systems) and will hopefully contribute towards the progression of applied titanium dioxide materials in water-splitting. The energetically unfavourable surface vacancy formation in rutile relative to anatase is made clear here. The role of Nitrogen doping in modifying the surface oxygen vacancy structures in anatase is examined with a focus on the significance in water splitting applications. This offers valuable information towards rational materials design.

The work is certainly worthy of dissemination and will be of value to numerous other researchers in the field of titania photocatalysts. I suggest several minor changes to raise the value of the information communicated in the manuscript:

We are very pleased with the positive and supportive comments from this Reviewer who has clearly recognised the novelty of our work and concluded the valuable contribution of this work towards rational materials design of titanium dioxide based materials for photocatalysis.

1. Much of the conclusions in the manuscript hinge upon the accuracy of lattice structure determined by Rietveld Refinement. In particular with biphasic mixtures, Rietveld refinements have a certain margin of error. Authors should make efforts to ascertain a possible margin of error regarding anisotropic lattice expansion found here and critically examine potential sources of experimental or methodological error in this measurement.

We totally agree with the need for the error estimations. As noted, we have performed a full analysis of error associated with the collection and refinement of the powder diffraction data. Error bars have now been added to the figures, as well as discussion into the clear difference in behaviour of the pristine and N-doped samples in the revised manuscript and supporting information.

2. Authors identify interstitial and substitutional nitrogen doping. If possible, this should be quantified, with some information regarding how one could feasibly control the ratio of interstitial and substitutional doping in an optimal way.

In light of the comment from this Reviewer, the XPS data has been further analysed to yield quantitative ratios for interstitial and substitutional nitrogen doping which are included in the Table 1 of the revised manuscript. Although it is beyond the scope of this study how one could feasibly control the ratio of interstitial and substitutional doping, we have included a small discussion and cited some important references for interested readers about this area (Lynch 2015, Zhang 2010) in the revised manuscript.

3. Some insights into the improvement of water splitting rate through materials design should be given.

We greatly appreciate the reviewer for this comment. The N-doped TiO₂ materials described in this manuscript have been characterised carefully with the help of various techniques. UV-vis spectra show that after N-doping the absorption range is broadened from UV to visible light (even near infrared) regime, which indicates that the N-TiO₂ can utilise wider range of the solar spectrum. Moreover, EPR results clearly show that N-doping is accompanied by oxygen vacancies formation which is generally considered as active site of oxygen evolution and may also enhance the oxygen mobility in the catalyst particles. With the help of time-resolved technique, we also observed the exciton lifetimes were substantially prolonged with the N inclusion. Further material characterization from our early reference (Nat. Commun. 2019, 10, 4421) has also been cited. In light of the comment from this Reviewer, we have discussed the materials design on the photocatalytic water splitting in the revised manuscript.

4. An explanation of whether enhanced performance is observed in biphasic mixtures would be welcome and if this can be mechanistically explained. In anatase rutile mixtures, improved performance is often reported to arise due to improved exciton separation by band bending mechanisms. Is a similar mechanisms predicted to occur in biphasic anatase / oxynitride phase assemblages?

In light of the useful comment from this Reviewer, discussion has been added on the role of the anatase-rutile band bending mechanism that is also anticipated to give beneficial catalytic performance for the nitrogen-doped samples. We have noted that the work functions of rutile and titanium nitride are similar; hence it is reasonable to assume that the band bending effects are also contributing to the both systems.

5. The role of oxygen vacancies on surface adsorption needs to be considered. The nature of water adsorption, molecular or dissociative is expected to be influenced by oxygen vacancies in doped an undoped materials. It is also possible that water adsorption stabilizes surface oxygen vacancies as opposed to sub-surface ones.

We thank this Reviewer for his/her useful comments. In our previous paper (Nat. Comm **10**, 4421 2019) and SI, experiments were carried out to investigate the regeneration of surface oxygen vacancies over our catalyst in water vapour and liquid water at elevated temperatures, respectively. According to the post-mortem analysis measured by EPR, it is apparent that the EPR signals increase in a similar way in the conditions with water vapour and liquid water as that in pure N₂, which suggests that the oxygen vacancies can still be regenerated at elevated temperatures under the photocatalytic water splitting conditions. However, we agree with this Reviewer that the presence of water can still affect the adsorptive properties of surface oxygen vacancies in the samples as well as their relative stability of the subsurface vacancies. Although we have not yet able to address the issue in this paper, we have included the references of literature (i.e. Chem. Sci., 2014, 5, 2256; Nat. Mat. 2017, 16, 461) and corresponding discussion in the SI to bring such important point to the awareness of the readers.

Review #2:

The authors Christopher Foo, Yiyang Li, Konstantin Lebedev, Tian-Yi Chen, Sarah J. Day, Chiu Tang, and Shik Chi Edman Tsang present a structural study of rutile, anatase and mixed phases like P25 particle with and without nitrogen-doping by variable-temperature synchrotron X-ray powder diffraction. They also do a theoretical study of the mentions systems.

The authors report an anisotropic thermal expansion and a cubic titanium oxynitride phase at high level of nitrogen dopant. These internal structures and other products of

the different positions that the doping element adopts in the network have also been described in the theoretical work of Morgade and others. [Computational Materials Science 127 (2017) 204–210] The work is interesting but presents some details that need to be considered to be corrected. In particular, in the theoretical part of it, I am concerned about the lack of use of any of the strategies available in the literature, such as the use of hybrid potentials or the Hubbard factor to correct the known difficulty of the DFT theory in the representation of highly correlated systems such as TiO₂. [The Journal of Chemical Physics 135, 054503 (2011), Results in Physics 6 (2016) 891–896 ref]. On the other hand, in the present work they do use the Van der Waals correction, whose use has not shown to be superior for the adequate representation of TiO₂.

We are greatly indebted to this Reviewer who pointed out the potential issues and the inaccuracy of the Van der Waals correction in our calculations. We agree with him that the use of hybrid potentials or the Hubbard factor to correct the known difficulty of the DFT theory in the representation of highly correlated systems. Indeed, there are a number of reports that used Hubbard correction factor to account for the TiO₂ based systems and drew a similar conclusion but with more accurate calculations than our method. As a result, we have reminded the readers about such treatments and cited the related references to them.

Another issue I find its not correct is when in the introduction part the authors write “Generally, it is thought that the extraction of neutral oxygen atoms, and the accompanying electron transfer to the Ti cationic band structure and formation of positively charged oxygen vacancies (holes), during light illumination may occur in both the bulk and on the surface of TiO₂ crystals”. Oxygen vacancies are in fact n-type dopants [M.V. Ganduglia-Pirovano et al./ Surface Science Reports 62 (2007) 219–270]

My appreciation is that once the corrections are made, the work deserves to be published.

We are grateful to this Reviewer who can see the merits and novelty of this paper and recommended publication of this work after modification. He/she was right to point out that the oxygen vacancies are regarded as the n-type of dopants to TiO₂ based systems. As a result, we have rectified the description in the Introduction and also corresponding discussion has been inserted on the DFT calculations in the revised manuscript.

Reviewer #3 (Remarks to the Author):

In this study, the authors carried out a structural study of rutile, anatase, mixed phase TiO₂ and their corresponding N-doped samples. The effect of calcination temperature

on the doping impurities, structural defects, vacancy concentration were investigated using VT-SXPD, DFT, etc. Anisotropic thermal expansion in lattice parameters was discovered, and the authors claimed the clarification of relationship between sub-surface oxygen vacancy, nitrogen-doping level and photocatalytic activity. Although the work describes interesting findings, I am not very confident that the findings are important enough to be of general interests to Nature Comm. readership.

N-doped TiO₂ has been extensively studied but this study failed to demonstrate any significant improvements of N-doped TiO₂ for overall water splitting. The authors carried the water splitting at elevated temperature and pressure condition in an autoclave photoreactor. Such photocatalytic reactivities observed under extreme conditions have little practical meaning. In addition, the importance of the experimental findings is not well justified for the following reasons.

We thank for the comment from this Reviewer who also agreed with other two reviewers that our study is interesting but he/she was unsure whether it is important or not. We understand his/her main concerns on the temperature (pressure) effect and any practical advantages of using elevated temperature in photocatalysis. It is because photocatalytic water splitting is normally carried out at room temperature, without input of energy to heat up the reactor, although, there are prior limited works including our works of using elevated temperatures in the literature.

We feel that these concerns may probably arise due to the fact that we did not include the sufficient background information in the original submission. This is the paper we intend to address on the structural changes of N-doped TiO₂ for water splitting and mechanism elucidation, followed by our earlier publication in the same journal mainly on the temperature effect and catalyst optimization in water splitting over the N-doped TiO₂ (reference 34. *Li, Y. et al. photocatalytic water splitting by N-TiO₂ on MgO (111) with exceptional quantum efficiencies at elevated temperatures. Nat. Commun. 10, 4421 2019*) and our recent invited review on potential advantages of photocatalysis at elevated temperatures (*Li and Tsang. recent progress and strategies for enhancing photocatalytic water splitting, Mat. Today Sustain. 9, 100032, 2020*).

In brief, we previous found that the hydrogen production activity over N-doped TiO₂ is 6746 μmol g⁻¹h⁻¹ at 270°C, which is substantially higher than the same catalyst and also most other photocatalytic systems at room temperature claimed in the literature. We postulated in these papers that at elevated temperature, the improvement in kinetics could be one of the main reasons to overcome the rate limiting regeneration of lattice oxygen vacancies from this system.

For potential practical applications, in those cited papers, we mentioned that they are reported configurations and prototypes of solar concentrators, such as parabolic cylinder reflectors (*Energy Environ. Sci., 2013, 6, 1983*) that can provide enhanced

light irradiation as well as temperature for small/medium size applications. In addition, we have also highlighted in the papers that some strategies for example, the recovery of heat from superheat steam and using a number of possible exothermic coupling reactions with H₂ to provide the heat required for the system at large scale, etc. There are also devised exploitation plans to address some practical issues for potential applications including the injection of separated H₂ from photo splitting at elevated temperature for decentralized domestic devices into natural gas pipeline in UK and some parts of Europe, etc for caloric use of this renewable fuel.

Besides, further preliminary study has also been initiated by replacing the liquid water with water vapour, which could be more controllable, easier to operate, possess lower heat capacity (therefore uses less energy to heat up), and can be operated at lower pressure for the same temperature, etc. In another word, substituting liquid water with water vapour in continuous flow could make this system more practical and feasible at elevated temperature. It is noteworthy that the visible-light-driven water splitting system clearly works well even with water vapour, and lower pressures of water vapour have been briefly studied (in SI of the Nat. Comm **10**, 4421 2019). We therefore hope that this structural study if published could stimulate further works to address some practical issues in photocatalytic splitting.

In light of the comment, we have created a new section for potential advantages in the the revised manuscript (SI) so the readers can appreciate the previous works and issues in using elevated temperature over the N-doped TiO₂.

Notice that it is unclear whether surface or lattice oxygen vacancy plays the key role in catalysis at the elevated temperature in the literature. But, in this paper, the in-depth structural study of rutile, anatase and mixed phases (P25 particle) with and without nitrogen-doping by variable-temperature synchrotron X-ray powder diffraction (VT-SXPD) give for the first time that the good correlations of the anisotropic thermal expansion in lattice parameters (influenced by sub-surface oxygen vacancy), nitrogen-doping level and photocatalytic activity with respect to temperature. At high level of nitrogen-doping of anatase, a new cubic titanium oxynitride phase is also identified, which gives important hints on the fundamental shift in absorption wavelength, leading to excellent photocatalysis in visible light regime.

1. There is no evidence of overall water splitting. Is there a stoichiometric evolution of O₂ gas? What about the possibility of N₂ evolution as a result of the N anion oxidation? Are N anions acting as a hole-scavenger for H₂ evolution? Prove that this reaction is photocatalytic by carrying out the multi-cycle photoreactions in the same batch of photocatalyst. Show if N₂ gas is evolved or not during the evolution of H₂.

We greatly appreciate the comment of this reviewer, and with reference to his/her comments, we have carried out a 5-cycle photocatalytic water splitting reaction test in

the same batch, of which the results are shown in the revised manuscript. As can be seen in the Fig. S5, hydrogen and oxygen are produced stoichiometrically in 2:1 ratio, which indicates that this reaction is photocatalytic overall water splitting reaction. Also, there was no sign of nitrogen formation from the nitride was observed in the careful GC analysis. We have also carried out a 50-hour long durability test as shown in Fig. S6 in the revised manuscript, which also shows the stable hydrogen and oxygen production in 2:1 ratio within experimental error for over 50 hours with absolutely no N_2 evolution. It is noted that N^{3-} electro-oxidation to N_2 despite its high concentration still requires significantly higher applied potential than that of OH^- to take place and the nitride oxidation can only be favourable in molten medium in the total absence of water (Journal of Nuclear Materials 344, 128–135, 2005). Thus, the possibility that the doped N anions act as hole-scavengers in the presence of water is not likely to happen.

2. The photocatalytic activity of N-TiO₂ for water splitting seems to be negligible at ambient temperature. What is the use of the water splitting under extreme conditions of high temperature and pressure? If you find some important information from this high temperature study, such knowledge should be demonstrated to improve the photocatalytic activity of N-TiO₂ under ambient condition, which is completely absent in this study.

In our previous paper (Nat. Comm **10**, 4421 2019) we reported that high temperature treatment must be required to regenerate the oxygen vacancies of N-TiO₂ (playing the important role in photocatalysis) otherwise the photo water splitting is too slow to take place under ambient conditions. It is therefore essential to do the catalysis at elevated temperature (to promote the rate limiting of oxygen vacancy regeneration). We have included the summary in the SI of this paper for the background information as follows:

Figure 1. EPR patterns of the N-doped TiO₂ photocatalysts. **a:** N-doped TiO₂P1 (freshly prepared) at different times. After being treated in NH₃ at 550 °C for 8 h, the EPR of the freshly made sample was measured immediately. More EPR spectra were also collected after the sample was exposed to air for 0.5 h, 1 h, 3 h and 24 h; **b:** deactivated P1 after re-calcination in N₂ at different temperatures; P1 quenched from **c:** liquid water and **d:** water vapour at different temperatures.

‘Oxygen vacancies can trap unpaired electrons from the semi-conductive oxide, which is detectable by electron paramagnetic resonance (EPR). Thus, EPR measurements were carried out after N-P25-550 was freshly prepared and exposed to air for different time ranges (Fig. 1a). Interestingly, after the sample was exposed to air at ambient conditions for 1.5 hours, 40% of the EPR signal, that is indicative of the presence of oxygen vacancies, gradually disappeared and after 24 hours, only 23% of the original signal remained (Fig. 1a). This is attributed to the fact that oxygen sources in the air (i.e. O₂ and H₂O), when in contact with the particle surface, may gradually replenish the oxygen vacancies and redistribute the electrons, approaching that of pristine TiO₂. This could explain the fact that N-doped TiO₂ or hydrogenated TiO₂ do not necessary show good photocatalytic water splitting activity at room temperature under visible light illumination in air even though the remaining oxygen defects in bulk (require to diffuse to surface but the rate is very slow) can exert strong visible light absorption.

However, we also noticed that after re-calcining the N-doped TiO₂ in a N₂ atmosphere at elevated temperatures, the EPR signals of these materials re-emerge and become even larger, implying that more oxygen vacancies are regenerated at elevated temperatures (Fig. 1b). Thus, the surface oxygen vacancies formed in N-doped TiO₂ are vulnerable to oxygen sources at room temperature, but at elevated temperatures, the faster subsequent reactions can regenerate them to sustain the surface photocatalytic processes.’

As stated, we understand the concern about the practicality of using elevated temperatures. Actually, in our previous study, we demonstrated that the high temperature and visible light irradiation can both be provided solely by light with the help of a light furnace (Nat. Commun. 2019, 10, 4421). Since the light intensity is much enhanced in such configuration, the photocatalytic overall water splitting reaction can also be achieved at 200 °C with a saturated vapour pressure of 20 bar, giving promising hydrogen evolution rate as well. Thus, solar concentrators, such as parabolic cylinder reflectors could provide enhanced light irradiation and temperature as discussed.

3. I do not see any clear reasoning how unusual anisotropic thermal expansion in lattice parameters is related with the photocatalytic activity. Such enhanced activity at higher temperature might be generally observed in other semiconductor photocatalysts as well. As shown in Figure S1, the pure TiO₂ and N-TiO₂ show the similar temperature dependence qualitatively. Is there any evidence that this phenomenon is unique to N-TiO₂ only? After further addressing the above points, this study should be better suited to a more specialized journal (material, catalysis, or physical chemistry).

We are sorry that the Reviewer had apparently missed the novelty of our work and were unable to follow the reasoning how the anisotropic thermal expansion in lattice is related to photocatalytic activity. This was probably due to the unclearness of our original manuscript without giving the previous background information.

As mentioned, it is not yet clear whether surface or lattice oxygen vacancy (sub surface) plays the key role in catalysis over the N-TiO₂ in the literature, especially during our high temperature in the regeneration of oxygen vacancies for sustained catalysis. In this following up paper, the in-depth structural study of rutile, anatase and mixed phases (P25 particle) with and without nitrogen-doping by variable-temperature synchrotron X-ray powder diffraction (VT-SXPD) give for the first time that the direct correlation of the anisotropic thermal expansion in lattice parameters to activity with respect to temperature.

In brief, the anisotropic thermal expansion of anatase has been observed at 150°C, namely that the *c*-axis exhibits increased thermal expansion while that of the *a*-axis decreases. This has been attributed to the increased weakening of the long Ti-O_{ax} bond. At 200°C the phenomenon is countered which is attributed to the Jahn-Teller contraction of 5-fold coordinated Ti³⁺ polyhedra formed **by the onset formation of oxygen vacancies in sub surface** (such diffraction method cannot measure the surface oxygen vacancies directly).

This onset of sub surface oxygen-vacancy formation in the asymmetric nitrogen-doped anatase phase at 200°C also correlates with our onset in activity as shown in Figure 5b. The activation barrier for hydrogen evolution from photocatalytic water splitting is determined to be 60.6 kJ mol⁻¹, which is in good agreement with the activation barrier for oxygen-vacancy formation of 62.5 kJ mol⁻¹, which we derived from variable-temperature EPR (Figure S8 and S9). Additionally, this formation energy is in good agreement with the reported activation energy for oxygen-vacancy mobility in anatase of 67.5 kJ mol⁻¹¹¹, which indicates that the formation of oxygen vacancies in anatase is limited by the diffusion of vacancies between the sub-surface and the surface.

The Reviewer was right to point out that pure TiO₂ and N-TiO₂ show the similar

temperature dependence, since the favourable formation sub-surface oxygen vacancies over TiO₂ based materials is anticipated (supported by the calculations). But, for nitrogen-doping in N-TiO₂, the inclusion of nitrogen atoms, either as an interstitial or substitutional dopant, can apparently increase the concentration of the sub surface oxygen vacancies and Ti³⁺, and also narrow the band gap resulting in photocatalytic water splitting by visible light. This can demonstrate the unique roles of N doping in the N-TiO₂ for the photocatalytic water splitting at elevated temperature.

In light of the comment from this Reviewer, we have substantially rewritten the discussion and conclusion with the above information in the revised manuscript in order for readers to see the correlations clearly.

REVIEWERS' COMMENTS

Reviewer #1 (Remarks to the Author):

The authors have made considerable and sincere efforts to remediate the shortcomings identified by reviewers. This work advances the field of titanium dioxide photocatalysis and will be of value to a variety of researchers worldwide. I am happy to see it published.

Dorian Hanaor

Reviewer #3 (Remarks to the Author):

The authors addressed the raised issues in a reasonable way but I still believe that the investigated topic is too specific to be of general interests and has little practical meaning.